# Validity, Reliability and Reproducibility of OctoBalance Test as Tool to Measure the Upper Limb Compared to Modified-Upper Quarter Y-Balance Test

**DOI:** 10.3390/ijerph18105057

**Published:** 2021-05-11

**Authors:** Álvaro Velarde-Sotres, Antonio Bores-Cerezal, Marcos Mecías-Calvo, Stefanía Carvajal-Altamiranda, Julio Calleja-González

**Affiliations:** 1Facultad de Ciencias de la Salud, Universidad Europea del Atlántico, 39011 Santander, Spain; alvaro.velarde@uneatlantico.es (Á.V.-S.); antonio.bores@uneatlantico.es (A.B.-C.); marcos.mecias@uneatlantico.es (M.M.-C.); stefania.carvajal@unini.org (S.C.-A.); 2Departamento de Salud, Universidad Internacional Iberoamericana, Campeche 24560, Mexico; 3Department of Physical Education and Sport, Faculty of Education and Sport, University of the Basque Country (UPV/EHU), 01007 Vitoria, Spain

**Keywords:** OctoBalance test, injury prevention, neuromuscular assessment, upper limb, functional tests

## Abstract

The articular evaluation of range of motion (ROM) is currently used to observe imbalance or limitations as possible risk factors or predispositions to suffer future injures. The main aim of this study is to verify the concurrent validity, reliability and reproducibility of the OctoBalance Test (OB) as a valid and reliable tool to measure articular ROM of the upper limb compared to the modified-Upper Quarter Y-Balance Test (mUQYBT). The twenty-five participants were male athletes. All of them were assessed with OB and mUQYBT in medial, superolateral, and inferolateral directions in both right and left arms with a three-minute break during these attempts. The process was repeated a second time with a week gap between measurements. Pearson correlation and linear logarithmic regression were used to examine the relationship between scores obtained with OB and mUQYBT. In order to verify the reliability, the intraclass correlation coefficient (ICC) was used (3.1). Concordance and reproducibility were assessed using Bland–Altman’s graph. A perfect correlation and an almost linear logarithmic regression (R^2^ = 0.97) were observed between both measurement systems, with values of 73.531 ± 21.226 cm in mUQYBT and 69.541 ± 16.330 cm in OB. The differences were minimal between week one and week two. The assessment with Bland’s graph showed the concordance and reproducibility of scores, showing the dispersion and the upper and lower limits. OB is shown as valid in comparison to the other test as a reliable and reproducible tool for the assessment of the articular ROM in the upper limb, and it could be used for the evaluation of injuries.

## 1. Introduction

Functional movement tests are currently used by sports scientists for the assessment of injuries [1], as they evaluate mobility and balance and allow us to observe asymmetries or functional imbalance as possible risk factors or predispositions to suffer future injuries; however, these tests must be valid, reliable, sensitive, and specific for the assessment of each athlete who is supposed to be valued [2].

Different studies [3] have shown the efficiency of the functional tests for the assessment of injuries, as they evaluate mobility and balance and they allow us to observe imbalance or limitations as possible risk factors or predispositions to suffer future injuries [4].

Therefore, the identification of high-risk sports and the most common and severe injuries is of great importance, giving priority to the sports which present a higher risk of injury in terms of frequency and severity. Thus, different sports are related to different kinds and types of injuries. Moreover, age, gender, and the type of activity (training/competition) are factors that influence the prevalence of injuries [4].

Thus, there are multiple types of injury, including direct injuries, the result of a sudden event, specific and identifiable; and injuries caused by excessive use or overuse, caused by repeated micro-injuries, without any specific cause which could be identifiable [5].

Likewise, the risk of injury is determined by two types of factors: internal (intrinsic) and external (extrinsic). The first is directly related to the individual characteristics of the athlete, while the latter is the environmental risk factors [4].

The aforementioned risk factors can be divided into modifiable and non-modifiable. Although non-modifiable risk factors, such as gender and age, are of interest, it is also important to study the factors which are potentially modifiable with training, such as strength, balance, or flexibility [4].

To this end, the articular evaluation of range of motion (ROM) is currently used to observe imbalance or limitations as possible risk factors or predispositions to suffer future injuries due to this cause [6,7]. ROM is the maximum angle described between two body segments with a reference plane which is realized by means of joints. ROM can also be considered as the flexion of the joints or as the degree of muscle contraction. ROM can be used as reference data to detect muscle asymmetries. ROM can be affected by lack of flexibility because the muscle chains work less efficiently and the load changes can lead to injury. In addition, ROM is affected by anatomical, biomechanical, biochemical, and neurophysiological factors. In this sense, functional movement tests are currently used by sports scientists for the evaluation of injuries, as they evaluate mobility and balance and allow us to observe asymmetries or functional imbalance.

Different tests of functional evaluation can be found in the scientific literature [3] that studies injuries in the lower body, as it is the body area where most injuries are produced in the practice of sport [8] and less in the upper body.

Among the tests, there is the Star Excursion Balance Test (SEBT), a dynamic postural control test, usually used for the lower body, which consists of eight lines on the ground with a gap of 45° from one another [9]. This test is considered the gold standard.

Another test used in the assessment is the Y-Balance Test (YBT) (Functional Movement Systems). It is a kit that assesses mobility in three directions, initially used for the record of measures of the lower body [10]; it was later validated for the upper body by Gorman et al. [11] and was given the name of Upper Quarter Y-Balance Test (UQYBT).

Recently, Cramer et al. [12], based on UQYBT and with the aim of reducing the expenses of acquisition of these tests, established a measurement protocol consisting of three laces on the ground in a Y-shape and three wooden blocks as the reference of the reached distance, called modified-Upper Quarter Y-Balance Test (mUQYBT). The reliability and concurrent validity of this test were shown with these trials. The results of the study showed a correlation of R^2^ = 0.96 between the mUQYBT and UQYBT [12].

Finally, there is a measurement system, the OctoBalance Test (OB), system Check your Motion^®^, which involves an octagonal platform and two measurement systems (retractable steel ruler tapes) as a valid and reliable tool [3] to assess the functional capacities of athletes and to identify weaknesses and asymmetries. In addition, it has been shown as a valid and reliable tool [3] for the measurement of ROM in the lower limb for which, and up to date as far as we know, there are no previous level studies or evidence which show the use of OB test for the upper limb. Thus, its verification and validation for these types of measures in the aforementioned body area are necessary.

Therefore, the aim of this study is to verify the concurrent validity, reliability and reproducibility of the OB test as a measurement tool of the articular ROM in the upper limb in comparison with the mUQYBT. Thus, as a hypothesis of the study, it is proposed that OB is shown as valid in comparison to the other test, a reliable and reproducible tool for the assessment of the articular ROM in the upper limb, and it could be used for the evaluation of injuries.

## 2. Materials and Methods

The research had an experimental approach. A total of 25 male participants took part in the study. The OB and the mUQYBT were used as assessment instruments for the study. The assessment process consisted of the performance of different movements with the right arm and left arm on the OB and mUQYBT platforms, measuring afterward the lengths reached. The complete process was repeated a second time, with a week gap between measurements in May 2018. The evaluation was carried out at the laboratory of the European University of the Atlantic, Santander, Spain.

### 2.1. Participants

Twenty-five participants participated in the assessment, all male athletes with a mean age of 21.3 ± 2.4 years old, height of 176.5 ± 7 cm, and body mass of 72.96 ± 7.79 kg. There were 23 right-handed participants and 2 left-handed participants, all who self-reported having an active life with 11.6 h of weekly physical activity (Table 1). For the evaluation of physical activity, the International Physical Activity Questionnaire (IPAQ) was used. Before the test, the athletes were familiarized with OB for 3 months.

Before starting the study, all participants gave their reported consent. That document was compulsory for every participant in the study and had to be filled out and signed before any test during the time of the research.

As criteria of inclusion in the study, it was considered that the subjects were completely healthy and suitably fit for the test. Criteria of exclusion from the study by which subjects who fulfilled one or more criteria were rejected included having suffered an injury, both new or old, in the upper body; taking any type of medication; and having practized 24 h before the test any training or other type of exercise which limited the subjects for the assessment, as in both cases ROM may have been affected in the joint of the upper limb.

Moreover, the study was evaluated from the beginning by a Research Ethics Committee of the European Atlantic University, Santander, Spain (code: ID16-IN-022).

It must be noted that the obtained data were treated with the greatest confidentiality and scientific rigor, their use restricted by the guidelines for research projects following the scientific method required in each case, complying with the Organic Law 15/1999 of the 13th of December on the Protection of Personal Data (OLPPD); the proceedings used respected the ethic criteria of the Responsible Committee of Human Experimentation (established by law 14/2007, published in the Spanish Official State Gazette, n° 159) and the Helsinki Statement of 2008, updated in Fortaleza, October 2013.

### 2.2. Material

The OB was used as an assessment tool, system Check your Motion^®^ (Figure 1), which involves an octagonal platform and 2 measurement systems (retractable steel ruler tapes), as a valid and reliable tool [3] to assess the functional capacities of athletes and identify weaknesses and asymmetries and to provide continuous feedback during the practice of corrective exercises.

The OB has been shown as a valid and reliable tool [3] for the measurement of ROM in the lower limb for which, and up to date as far as we know, there are no previous level studies or evidence which show the use of the OB test for the upper limb. Thus, its verification and validation for these types of measures in the aforementioned body area compared to the mUQYBT are necessary. This test is considered the gold standard.

At the same time, the mUQYBT was used (Figure 2); it consists of 3 laces on the ground in a Y shape and 3 wooden blocks as the reference of the reached distance. This measurement test is an easy, effective, and scientifically valid way [12] to assess the functional capacities of athletes and to identify weaknesses and asymmetries.

For the test, flexible tapes were fixed on the ground with 5 cm wide transparent adhesive tape. The tapes formed a Y shape with two angles of 135° and 90°, respectively, measured with a goniometer [12]. A mark was put in the intersection of the three lines as a reference point to rest the hand. The calibration of the tapes was performed in comparison with an approved tape for athletic competitions (Polanik, Poland). Three wooden blocks, 2 × 4 × 8 cm^3^, were made that replicated the assessment by Cramer et al. [12].

Finally, to collect the data, a record sheet was designed ad hoc and was used as a guide for the established development of the measurement protocol, the collection of personal data, and the description of the sample.

### 2.3. Testers

Two testers (Á.V.-S. and A.B.-C.) were used in the assessment. Two experienced men, both in practice and theory, were in charge of the measurement, observation, and data collection. Both testers were sports scientists with a master’s degree in prevention and recovery of sports injuries as well as in high-performance sports. In addition, the intraclass correlation coefficient (ICC) was calculated. The ICC (3.1) was used in which each assessor assesses each item, but the assessors are the only assessors of interest. The two testers were blinded to the collected measures during the trials and the data collected by the other one (J.C.-G.). Thus, all subjects were assessed by both testers, which allowed a much more specific assessment.

### 2.4. Protocol

The assessment process consisted of the performance of different movements in the medial (M) (Figure 3), superolateral (SL) (Figure 4), and inferolateral (IL) (Figure 5) directions with the right arm (R) and left arm (L) on the OB and mUQYBT platforms, measuring afterward the length reached.

Before the test, the athletes were familiarized with OB for 3 months; moreover, they had to fill out a form with their personal data, including their name and surname, gender, age, body mass, height, sport practized, previous injuries in case there were any, and experience (years) in the sport. Simultaneously, information about dominance in the upper limb was requested; twenty-three of the subjects were right-handed and two subjects left-handed.

Before starting the assessment, there was a warm-up which consisted of 3 series of 10 repetitions of a squat without external load, push-ups (arms) without any load, and lunges without any load, all of them with a 60-second break among series. Furthermore, there was not any stretch before the tests and all subjects were informed about the necessity of correct performance, avoiding compensations and flexions of the elbow.

For the assessment, a measurement of the length of the upper limb was carried out in which the subjects were standing with 90° abduction of the shoulder, with a fully extended elbow and hand in a neutral position, from vertebra C7 to the longest finger in the hand, with a mean result of 89.64 ± 3.83 cm for the right arm and 90.4 ± 3.65 cm for the left arm.

The mUQYBT and OB tests were carried out following the instructions of UQYBT [11]. All the participants were given a demonstration of the performance of the test to be conducted. Firstly, mUQYBT was carried out, and once finished, the OB test was performed with the same protocol until all measurements of each test were completed.

The test was performed with the participants barefoot in order to avoid any stability or balance provided by any footwear. Following the recommendations by Gonzalo-Skok et al. [3], three attempts were performed for each direction: medial, superolateral, and inferolateral, with each arm having a 3 min break during these attempts.

All the processes were repeated a second time, with a week gap between measurements, in May 2018. All the subjects were assessed by the same testers (Table 2 and Table 3).

The attempts were not considered valid if the stability failed, the subject moved or did not touch the platform, there was no contact with the wooden block or contact was made incorrectly (with the tip of the fingers), the feet rose from the ground, or they moved [11,12]. This way, when the athlete performed a repetition that was considered null, the test was repeated again until it was performed correctly.

Due to the structure of the platform, 18 cm were added to the different measures obtained with the OB, which correspond to the distance from the centre to the edge of the platform and which allowed the correct measurement with the retractable steel ruler tapes.

Before the correlations in performance were carried out, all the scores of each subject were transformed in order to individualize them and obtain a more specific mean [12]. Therefore, the distance from the vertebra C7 to the tip of the longest finger of the limb was multiplied by the obtained values in each of the tests and divided by 100 [13]. Afterward, the mean of the scores of mUQYBT and OB was calculated, and this mean was the value used to carry out the correlation in the performance test. Moreover, in order to establish the correlation, the results obtained in the first week were taken into account. The results identified 25 participants with 73.53 ± 21.22 cm in mUQYBT and 69.54 ± 16.33 cm in OB in the medial, superolateral, and inferolateral directions of the right and left arms in week 1. The assessment of R^2^ showed near correlation and an almost linear logarithmic regression between the scores of OB and mUQYBT in week 1, with an adjustment value of R^2^ = 0.97 for the logarithmic values of the mean.

### 2.5. Statistical Analysis

Data are shown as mean (±) standard deviation (SD). Firstly, and after segmenting the test, an assessment was carried out to verify the normal distribution of the data. Shapiro–Wilk tests (<50) were applied in order to verify the normality of data. The assessed variables showed a normal distribution as we opted for the use of parametric tests. A significance level of *p* ≤ 0.05 was accepted, with confidence intervals (CI) of 95% in all the measures.

In order to verify the validity of OB, Pearson correlation was used to examine the relationship between scores obtained with OB and mUQYBT in week 1. The correlation between both tests, OB and mUQYBT, was carried out by the Pearson correlation (r) of the means, with a confidence interval of 95%, where the result of the correlation is shown between 0 and 1 (r: <0.1, trivial; >0.1–0.3, small; >0.3–0.5, moderate; >0.5–0.7, large; >0.7–0.9, very large; and >0.9–1.0, almost perfect) [14]. The R^2^ was calculated with a statistical linear trend line using logarithmic values of the means in order to get a greater adjustment of the sample.

In order to verify the reliability of OB, the ICC was calculated, and ICC was used (3.1) [15]. The ICC (3.1) was used in which each assessor assesses each item, but the assessors are the only assessors of interest. In addition, reliability was calculated from a single measurement.

Likewise, the reproducibility of the tests was verified through the assessment of the different means of the OB between week 1 and week 2 and the means of mUQYBT between week 1 and week 2.

Finally, concordance and reproducibility were assessed using Bland–Altman’s graph [16], a graphical method that allows us to compare 2 measurement techniques on the same quantitative variable and to carry out an assessment of the differences between the means.

The Bland–Altman graph was made with the scores of OB and mUQYBT in the medial, superolateral, and inferolateral directions of both right and left arms during week 1. The estimate of the difference was carried out with confidence intervals (IC) of 95% which result from [mean difference + 1.96 X, the standard error of the differences] [16].

All the assessments were carried out using version 25 of Statistics Software package^R^ (SPSS^®^ Inc., Chicago, IL, USA) except the graph of logarithmic correlation, which was made with the Office Excel Package 2016.

## 3. Results

The statistics assessment identified 25 participants with 73.53 ± 21.22 cm in mUQYBT and 69.54 ± 16.33 cm in OB in the medial, superolateral, and inferolateral directions of the right and left arms in week 1 (Table 4).

Variables of mUQYBT in week 1 obtained values of 99.87 ± 12.93 cm (right medial), 101.51 ± 12.03 cm (left medial), 56.41 ± 11.40 cm (right superolateral), 57.80 ± 10.26 cm (left superolateral), 64.24 ± 11.50 cm (right inferolateral), and 61.32 ± 7.88 cm (left inferolateral) (Table 5).

Variables of OB in week 1 obtained values of 89.95 ± 8.42 cm (right medial), 88.38 ± 8.99 cm (left medial), 53.26 ± 9.89 cm (right superolateral), 52.97 ± 9.09 cm (left superolateral), 67.23 ± 11.75 cm (right inferolateral), and 65.43 ± 10.39 cm (left inferolateral) (Table 6).

The assessment of R^2^ (coefficient of determination) showed near correlation and an almost linear logarithmic regression [14] between the scores of OB and mUQYBT in week 1, with an adjustment value of R^2^ = 0.97 for the logarithmic values of the mean (Figure 6).

The ICC (3.1) showed high reliability [17] with respect to the means obtained in the comparisons, obtaining 0.971 Cronbach’s Alpha coefficient based on the mean of the correlations in the variables, which showed a high internal consistency of OB and mUQYBT in week 1 [18] (Table 7).

On the other hand, the results show the reproducibility in week 1 and 2 in both mUQYBT (81.63 ± 23.57 cm vs. 77.90 ± 22.92 cm) and OB (56.45 ± 19.74 vs. 56.14 ± 17.87), indicating that the differences were minimal (Table 8).

Finally, the results show the concordance and reproducibility related to the scores of OB and mUQYBT in the right medial direction (Figure 7), left medial (Figure 8), right superolateral (Figure 9), left superolateral (Figure 10), right inferolateral (Figure 11), and left inferolateral (Figure 12) in week 1. These data indicate the degree of dispersion which is determined by the range of the differences in the results of the two methods. Moreover, they show the upper and lower limits of concordance with confidence intervals of 95% [16].

## 4. Discussion

The aim of this study was to verify the concurrent validity, reliability and reproducibility of OB as a valid and reliable tool to measure articular ROM of the upper limb compared to mUQYBT. We can indicate that the OB is shown as a valid, reliable, and reproducible tool for the assessment of the articular ROM in the upper limb, and it could be used by sports scientists for the evaluation of injuries.

Thus, the data obtained with OB (69.54 ± 16.33 cm) in the medial, superolateral, and inferolateral directions of both right and left arms show the efficiency of the functional tests [3] for the assessment of injuries, as they assess mobility and balance and allow us to observe imbalance or limitations as possible risk factors or predispositions for future injuries.

Likewise, the results of mUQYBT (73.53 ± 21.22 cm) show the validity and reliability of the test [12] in the medial, superolateral, and inferolateral directions of both right and left arms, as a test of functional assessment to observe imbalance or limitations.

Therefore, the coefficient of correlation R^2^ = 0.97 shows the real validity of OB, indicating a high probability that the same result is obtained using OB and mUQYBT, which demonstrates the predictive value of the test as well as the relationship between both assessment tools since R^2^ = 1 is a perfect linear adjustment [14]. Therefore, our results were similar to those of Cramer et al. [12], which showed correlations of R^2^ = 0.96 between the mUQYBT and UQYBT. Thus, the results in the mUQYBT and the OB show the efficiency of the assessment of the articular range of motion to observe imbalance or limitations as possible risk factors or predispositions to suffer future injuries [6,7]; therefore, the use of these tools and assessments is essential.

In this regard, it can be stated that sports success depends on many and different factors, which stand out more than the athlete and the stimuli of the training applied by the coach. Therefore, early identification of the injury and the interventions based on scientific evidence are the key factors for the prevention and treatment of sports injuries, but this is only possible with an assessment system that is reliable and valid, such as the one used [19,20].

Moreover, the intraclass correlation coefficient (ICC) (3.1) shows the high reliability of OB [17], which is in line with other assessments that use this ICC (3.1), such as that of Plisky et al. [10], in order to measure the reliability and performance of UQYBT. Thus, the mean of the correlations of the variables shows high reliability related to the obtained measures in the comparisons of OB and mUQYBT, obtaining 0.971 Cronbach’s Alpha coefficient, which shows the high consistency of the used tests [18]. Therefore, it can be verified that these assessment tools are reliable for their use in rehabilitation [17].

Likewise, the reproducibility of OB is shown in the minimal differences of the results in comparing week 1 and week 2 (56.45 ± 19.74 vs. 56.14 ± 17.87); such monitoring is essential for assisting athletes in knowing their evolution or progress and establishing at the same time a bond with their trainer or sports scientist in order to reduce the risk of injury [4]. The results also show the reproducibility of mUQYBT (81.63 ± 23.57 cm vs. 77.90 ± 22.92 cm) in week 1 and week 2.

Finally, the Bland and Altman graph shows the concordance and reproducibility of the scores of OB and mUQYBT in different directions, obtaining similar results in both tests. In this sense, similar studies [12] show this same concordance and reproducibility between the mUQYBT and the UQYBT, reflecting the degree of dispersion, which is established by the range of differences in the results of the methods and the upper and lower limits of concordance with confidence intervals of 95% [16]. Thus, the validity of OB related to mUQYBT is shown, as the measures are replicated with a minimal difference, which provides sports scientists with a very useful tool for the assessment of articular ROM in the upper limb.

The type of sport, the sample, and the experience, as it usually happens in these kinds of assessments, can be considered as limitations, as the election of some implies the rejection of others, which could also provide other types of data of wide interest. On the other hand, it provided a very homogeneous sample including only men with longitudinal monitoring for the study and the collection of results.

It should be pointed out that the conclusions provided have been carried out according to the results obtained in our research and selected under our eligibility criteria, using valid and reliable tools to identify the validity, reliability and reproducibility of the OB as an assessment tool of the articular ROM in the upper limb.

The tests of functional assessment are currently used by sports scientists for the assessment and prevention of injuries, but this type of test must be valid, reliable, sensitive, and specific for the assessment of each type of athlete who is supposed to be valued, as this research shows.

Thus, more articles studying injuries in the lower body can be found in the scientific literature; these are more plentiful than those studying the upper body, as the lower body area is where most injuries are produced when practizing sport and less in the upper body.

Therefore, the verification of the OB test would be necessary for the detection of asymmetries in the upper limbs of athletes, depending on the moment of the season, in order to assess the mobility and balance, allowing us to observe imbalance or limitations as possible risk factors or predispositions to suffer future injuries.

## 5. Conclusions

This research shows the validity of the OB (R^2^ = 0.97) as an assessment tool compared to the mUQYBT as well as its reliability and reproducibility, as it is able to replicate measures with a minimal difference, which provides sports scientists with a useful tool to assess articular ROM in the upper limb.

Therefore, with the OB, it is possible to use these measures in the medial, superolateral, and inferolateral directions in order to assess mobility and balance, which allows us to observe imbalance or limitations as possible risk factors or predispositions to suffer future injuries.

Therefore, we can claim that the OB is a valid, reliable, and reproducible tool for the assessment of articular ROM in the upper limb which can be used by sports scientists for the assessment and prevention of injuries.

## Figures and Tables

**Figure 1 ijerph-18-05057-f001:**
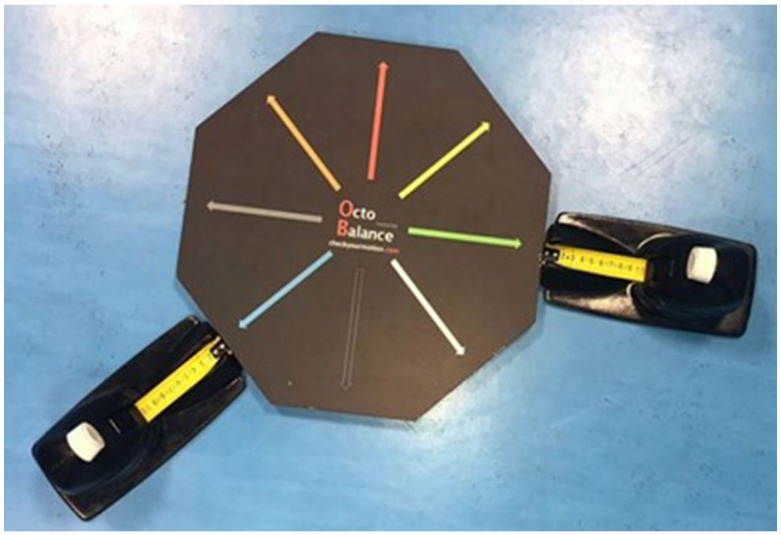
OctoBalance Test.

**Figure 2 ijerph-18-05057-f002:**
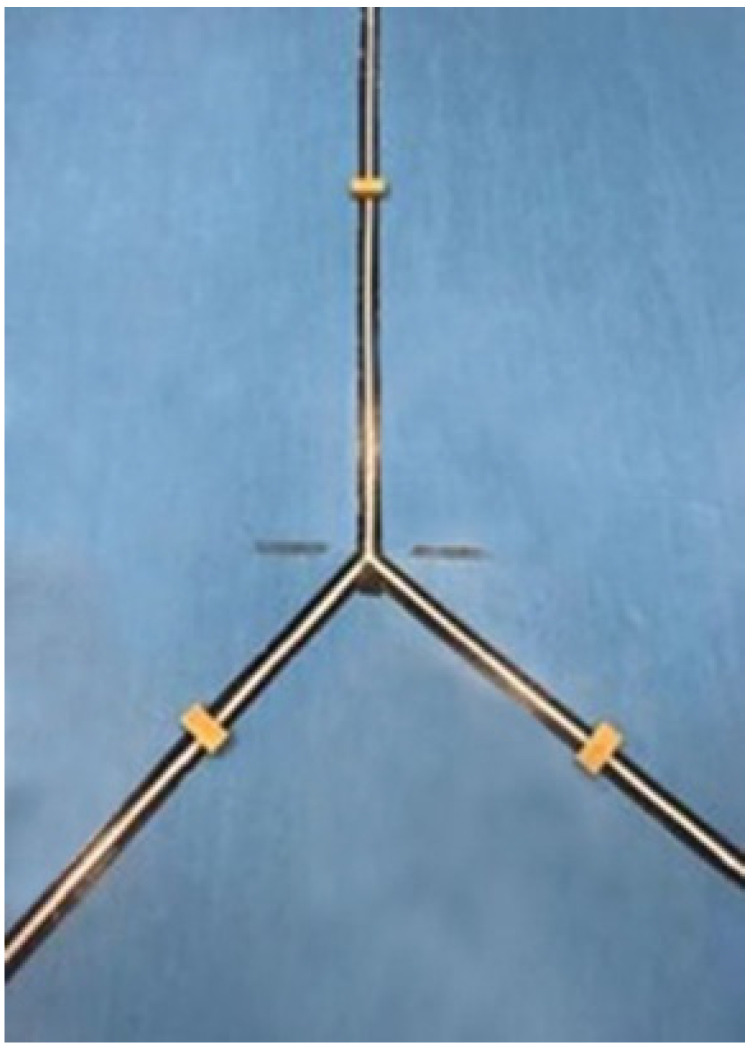
Modified-Upper Quarter Y-Balance Test (mUQYBT).

**Figure 3 ijerph-18-05057-f003:**
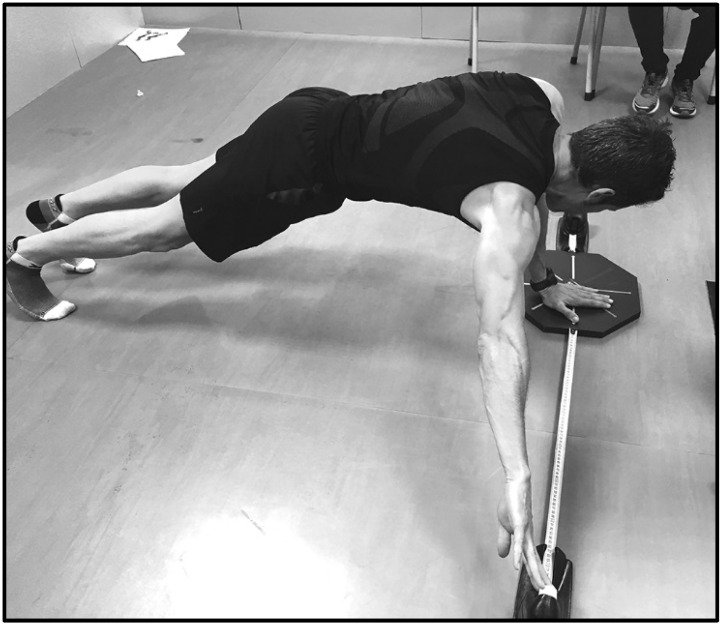
OB in the medial direction.

**Figure 4 ijerph-18-05057-f004:**
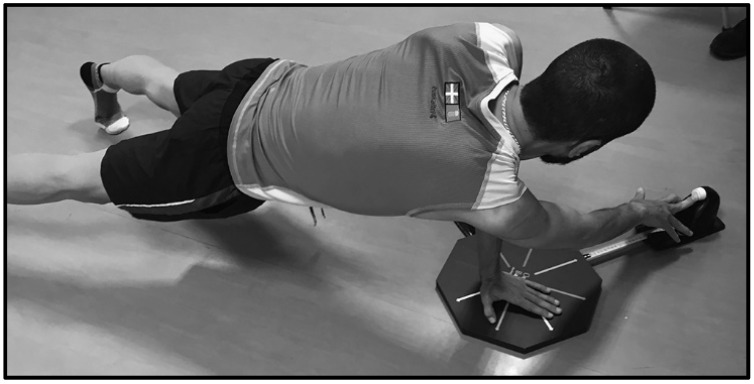
OB in the superolateral direction.

**Figure 5 ijerph-18-05057-f005:**
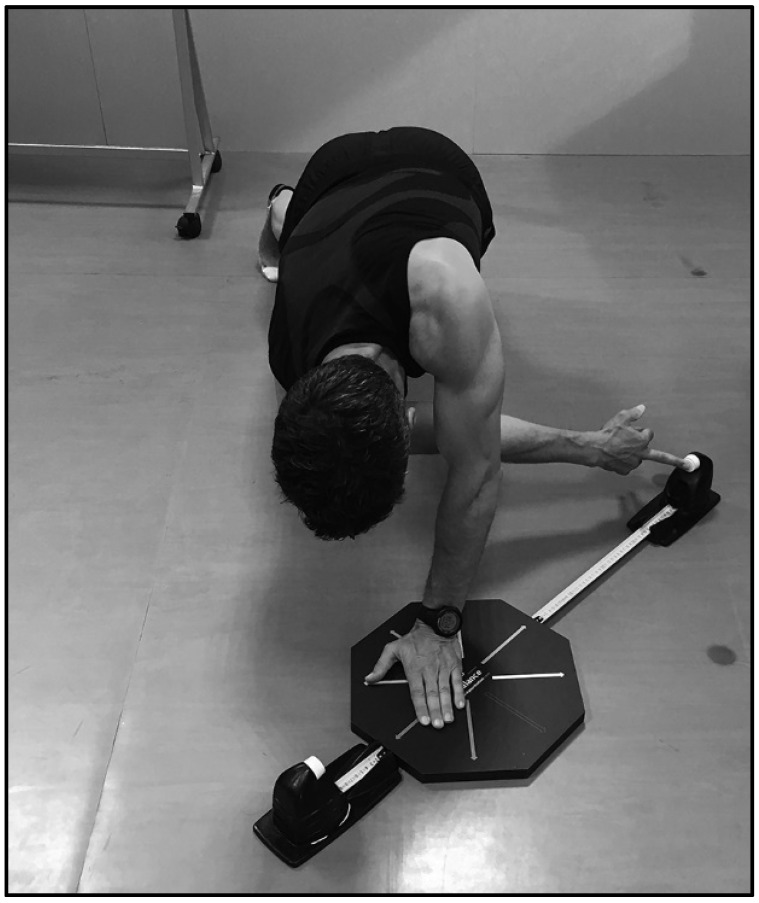
OB in the inferolateral direction.

**Figure 6 ijerph-18-05057-f006:**
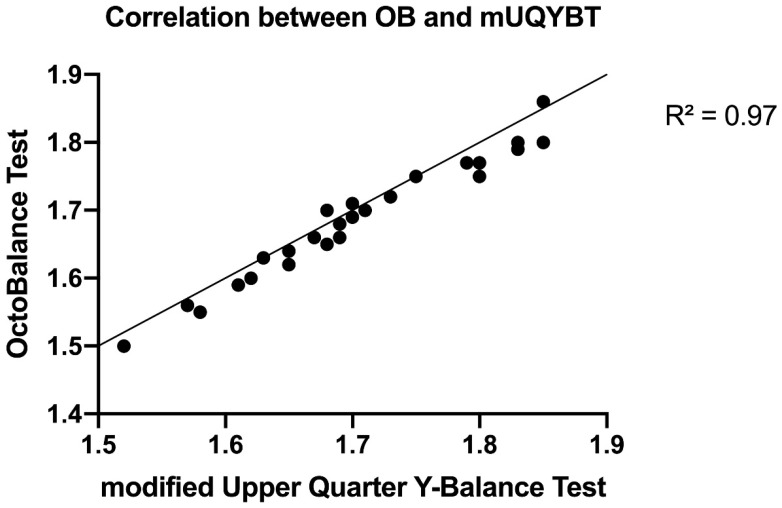
Logarithmic correlation between OB (cm) and mUQYBT (cm).

**Figure 7 ijerph-18-05057-f007:**
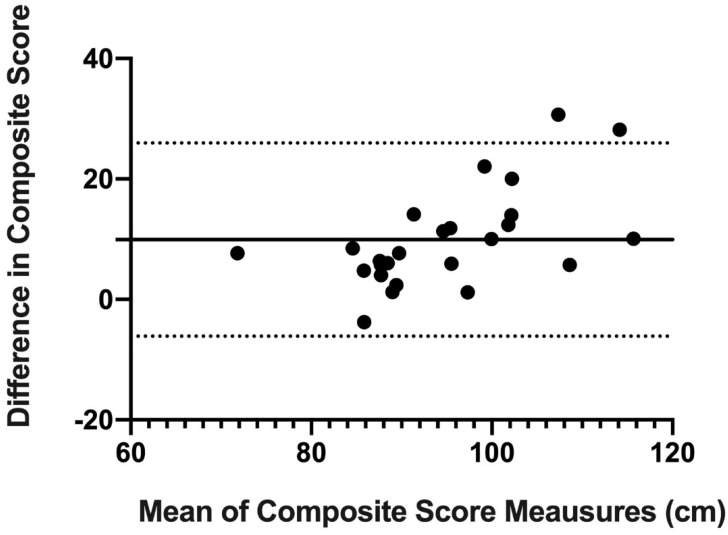
Bland–Altman right medial.

**Figure 8 ijerph-18-05057-f008:**
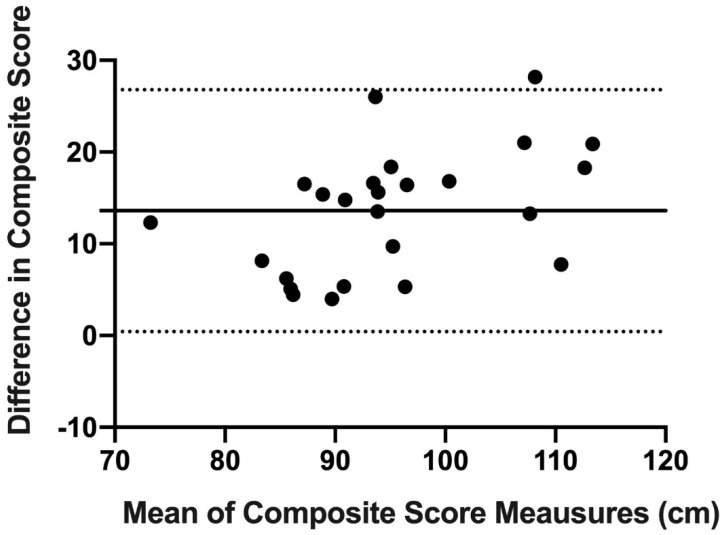
Bland–Altman left medial.

**Figure 9 ijerph-18-05057-f009:**
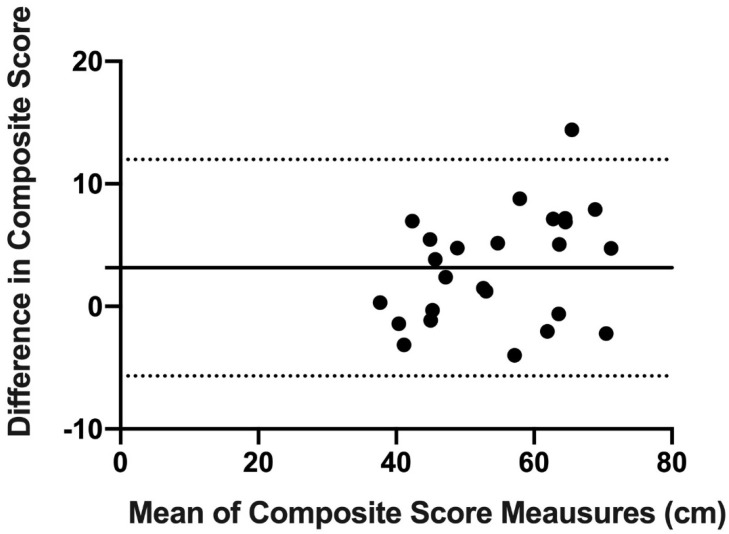
Bland–Altman right superolateral.

**Figure 10 ijerph-18-05057-f010:**
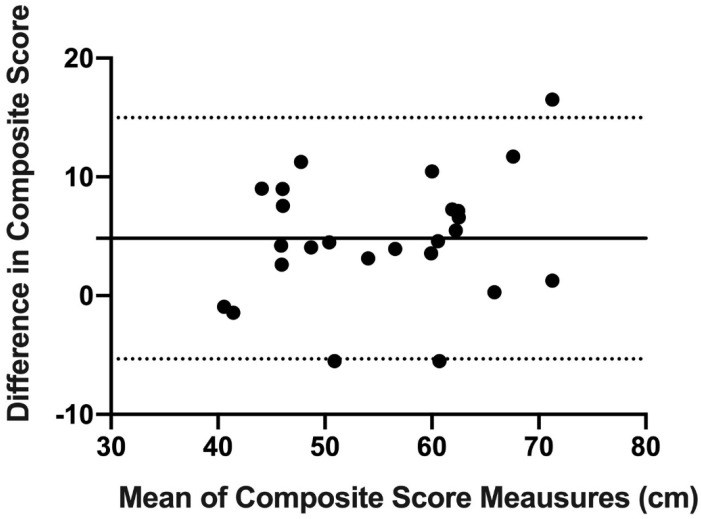
Bland–Altman left superolateral.

**Figure 11 ijerph-18-05057-f011:**
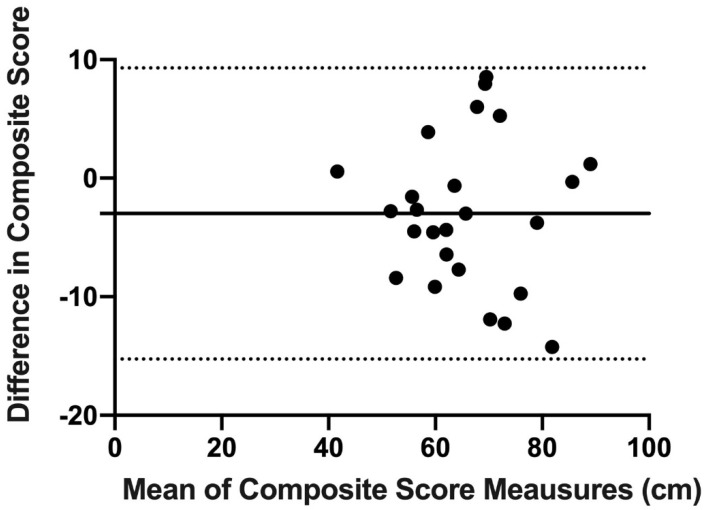
Bland–Altman right inferolateral.

**Figure 12 ijerph-18-05057-f012:**
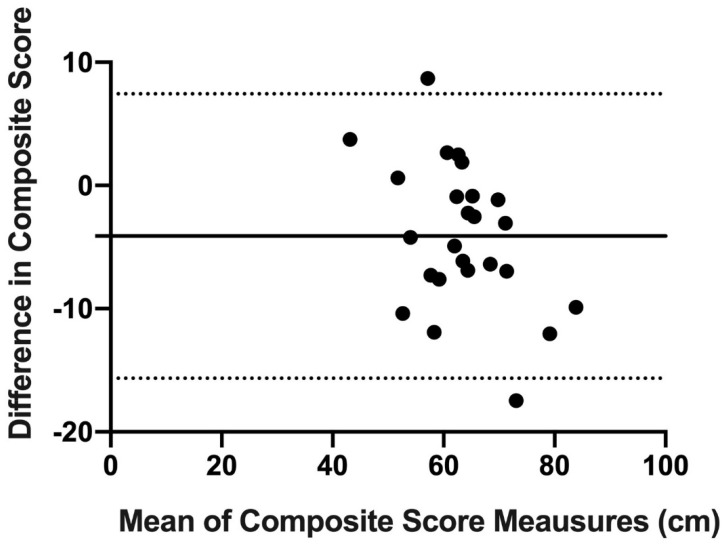
Bland–Altman left inferolateral.

**Table 1 ijerph-18-05057-t001:** Characteristics of the participants in the assessment.

	Age (years)	Height (cm)	Body Mass (kg)
N	25	25	25
Mean	21.3	176.5	72.96
Standard deviation	2.4	7	7.79

**Table 2 ijerph-18-05057-t002:** Chronogram of the mUQYBT measurements.

Modified-Upper Quarter Y-Balance Test																						
*Right-1*			*Right-2*				*Right-3*				*Left-1*				*Left-2*				*Left-3*			
M	SL	IL	**B**	M	SL	IL	**B**	M	SL	IL	**B**	M	SL	IL	**B**	M	SL	IL	**B**	M	SL	IL
30″			**3′**	30″			3′	30″			**3′**	30″			**3′**	30′′			**3′**	30″		

M: medial; SL: superolateral; IL: inferolateral; B: break between measurements.

**Table 3 ijerph-18-05057-t003:** Chronogram of the OB measurements.

OctoBalance Test																							
*Right-1*				*Right-2*				*Right-3*				*Left-1*				*Left-2*				*Left-3*			
**B**	M	SL	IL	**B**	M	SL	IL	**B**	M	SL	IL	**B**	M	SL	IL	**B**	M	SL	IL	**B**	M	SL	IL
**3′**	30″			**3′**	30″			**3′**	30″			**3′**	30″			**3′**	30″			**3′**	30′′		

M: medial; SL: superolateral; IL: inferolateral; B: break between measurements.

**Table 4 ijerph-18-05057-t004:** Modified-Upper Quarter Y-Balance test and OctoBalance test week 1.

	N	Minimum	Maximum	Mean	Std. Deviation
mUQYBT	25	56.41	101.52	73.53	21.22
OB	25	52.98	89.96	69.54	16.33

**Table 5 ijerph-18-05057-t005:** Values of Modified-Upper Quarter Y-Balance Test in week 1.

mUQYBT	Mean	Std. Deviation	N
Right Medial	99.87	12.93	25
Left Medial	101.51	12.03	25
Right Superolateral	56.41	11.40	25
Left Superolateral	57.80	10.26	25
Right Inferolateral	64.24	11.50	25
Left Inferolateral	61.32	7.88	25

**Table 6 ijerph-18-05057-t006:** Values of OctoBalance test in week 1.

OB	Mean	Std. Deviation	N
Right Medial	89.95	8.42	25
Left Medial	88.38	8.99	25
Right Superolateral	53.26	9.89	25
Left Superolateral	52.97	9.09	25
Right Inferolateral	67.23	11.75	25
Left Inferolateral	65.43	10.39	25

**Table 7 ijerph-18-05057-t007:** Intraclass Correlation Coefficient.

	Intraclass Correlation	95% Confidence Interval		Sig
		Lower Bound	Upper Bound	
Single Measures	0.735	0.615	0.847	0.000
Mean Measures	0.971	0.950	0.985	0.000

**Table 8 ijerph-18-05057-t008:** Reproducibility of mUQYBT and OB in week 1 and week 2.

	Minimum	Maximum	Mean	Std. Deviation
mUQYBT 1	62.81	112.24	81.63	23.57
mUQYBT 2	57.77	107.16	77.90	22.92
OB1	40.53	82.31	56.45	19.74
OB2	37.99	78.91	56.14	17.87

## Data Availability

The data presented in this study are available on request from the corresponding author. The data are not publicly available due to privacy.

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
