# Peer review of "Validity, Reliability and Reproducibility of OctoBalance Test as Tool to Measure the Upper Limb Compared to Modified-Upper Quarter Y-Balance Test"

_ijerph, 2021, doi:10.3390/ijerph18105057_

Round 1

Reviewer 1 Report

ABSTRACT

Lines 15-17: the background seems not to be in line with the aim of the study. The aim was to test the OctoBalance as a measure for ROM. Thus, the background does not provide information on that.

Line 17: concurrent validity instead only “validity” as the aim of the study

Line 20: specify the population (players?)

Lines 20-21: please describe the study design. The two weeks was to within-test analysis?

Line 31: valid in comparison to the other test.

INTRODUCTION

Line 42: prevention injuries? How?

Lines 42-44: the paragraph should be expanded about the concept of “functional movement”, the type of “functions” measured and the specificity of these tests.

Lines 45-48: reinforce the use of assessment as “prevention injury” is not a clear approach. Assessments may help to identify the status of the player, provide feedback for sports training but it is hard to believe that a test “itself” is a solution for prevention injury.

Lines 42-63: I suggest a deep revision of these paragraphs. Multidimensional factors affecting injuries should be considered:

Maybe some good reads:

McCunn, R., aus der Fünten, K., Fullagar, H. H., McKeown, I., & Meyer, T. (2016). Reliability and association with injury of movement screens: a critical review. Sports medicine46(6), 763-781.

Kalkhoven, J. T., Watsford, M. L., & Impellizzeri, F. M. (2020). A conceptual model and detailed framework for stress-related, strain-related, and overuse athletic injury. Journal of science and medicine in sport.

Stern, B. D., Hegedus, E. J., & Lai, Y. C. (2020). Injury prediction as a non-linear system. Physical therapy in sport41, 43-48.

Line 64 defines the concept of “range-of-motion”. Describe how can be used to “observe imbalance or limitations”

Line 69-70. Why SEBT? Is it the gold standard for that? Maybe not. A stronger rationale should be written. Describe the options, and then, move to the SEBT.

Lines 71-82: merely descriptive. Add information about the accuracy, precision, and repeatability of these tests (with values). Additionally, describe the outcomes and how the outcomes have been analyzed in the literature. Add information about the usability of these tests in scientific research.

Line 83. The introduction to the OB test is not clear. Why this test? What difference in comparison with the others? How he has been used?

Line 88: validity? Or concurrent validity?

METHODS

Line 91: start the section with an “experimental approach”. Present key elements of study design early in the paper. Describe the setting, locations, and relevant dates, including periods of recruitment, exposure, follow-up, and data collection. Add the timeline of the study.

Line 93: add information about the “participants”, namely the population, past in sports. Also, add information if they had experienced in the test since this can be a bias for the learning curve.

Line 147: the observers were tested for intra- and inter-observers reliability?

Line 168: for 3 months?

Line 215: Add the outcome extracted and explain how quantitative variables were handled in the analyses.

Line 233: details the options to perform the ICC.

RESULTS

Figure 6: add the units in x and y axes. The quality of the figure must be better (resolution).

Table 7. replace “,” by “.” In the sig

Figures 7-12: resolution must be better, the numbers must be bigger and the y and x-axes must have the caption.

DISCUSSION

The discussion should be strongly re-written. The writing must be about the concurrent validity and reliability of the test. Most of the discussion is cantered in something that was not tested: injury prevention, injury risk, imbalances. None of the topics were analyzed in this study. Examples:

Line 306: add the main findings and answers to the research question.

Line 309: “for prevention of injuries”? “allow observing imbalance or limitations”? This was a study?

Lines 314: It was the aim of the study? Was tested?

Lines 320-322: it was studied? The results were found in this experimental approach? Predicting injury?

Suggestions: Give a cautious overall interpretation of results considering objectives, limitations, multiplicity of analyses, results from similar studies, and other relevant evidence. Discuss the generalisability (external validity) of the study results. Discuss limitations of the study, taking into account sources of potential bias or imprecision. Discuss both direction and magnitude of any potential bias

Author Response

# Editor: Lines 15-17: the background seems not to be in line with the aim of the study. The aim was to test the OctoBalance as a measure for ROM. Thus, the background does not provide information on that.

# Author’s response: Agree, good detail. Thanks so much. We have changed everything. Yellow in the test.

“The articular evaluation Range of Motion (ROM) is currently used to observe imbalance or limi-tations, as possible risk factor or predisposition to suffer future injures”.

# Editor: Line 17: concurrent validity instead only “validity” as the aim of the study.

# Author’s response: Thanks so much. We have changed it. Yellow in the test.

# Editor: Line 20: specify the population (players?)

# Author’s response: Agree, good detail. Thanks so much. We have changed it. Yellow in the test. “Male athletes”

# Editor: Lines 20-21: please describe the study design. The two weeks was to within-test analysis?

# Author’s response: Agree. Thanks so much. We have changed it. Yellow in the test.

Twenty-five participants, male athletes, all of them assessed with OB and mUQYBT, in medial, superolateral and inferolateral directions, both right and left arms, having a 3-minute break during these attempts. All the process was repeated a second time, with a week gap between measurements”

# Editor: Line 31: valid in comparison to the other test.

# Author’s response: Agree, good detail. Thanks so much. We have changed it. Yellow in the test.

“OB is shown as a valid in comparison to the other test, reliable and reproducible tool for the as-sessment of the articular ROM in the upper limb, and it could be used for the evaluation and prevention of injuries”.

# Editor: Line 42: prevention injuries? How?

# Author’s response: Agree. Thanks so much. We have changed it. Yellow in the test.

“Functional movement tests are currently used by sports scientists for the for the as-sessment of injuries , as they evaluate the mobility and balance and allow us to observe asymmetries or functional imbalance, as possible risk factor or predisposition to suffer future injuries, but these tests must be valid, reliable, sensitive and specific for the assess-ment of each athlete who is supposed to be valued”.

# Editor: Lines 42-44: the paragraph should be expanded about the concept of “functional movement”, the type of “functions” measured and the specificity of these tests.

# Author’s response: Agree. Thanks so much. We have changed it. Yellow in the test.

 “Functional movement tests are currently used by sports scientists for the for the as-sessment of injuries, as they evaluate the mobility and balance and allow us to observe asymmetries or functional imbalance, as possible risk factor or predisposition to suffer future injuries, but these tests must be valid, reliable, sensitive and specific for the assess-ment of each athlete who is supposed to be valued”

# Editor: Lines 45-48: reinforce the use of assessment as “prevention injury” is not a clear approach. Assessments may help to identify the status of the player, provide feedback for sports training but it is hard to believe that a test “itself” is a solution for prevention injury.

# Author’s response: Agree. Thanks so much. We have changed it. Yellow in the test.

# Editor: Lines 42-63: I suggest a deep revision of these paragraphs. Multidimensional factors affecting injuries should be considered:

# Author’s response: Agree. Thanks so much. We have changed it. Yellow in the test.

# Editor: Line 64 defines the concept of “range-of-motion”. Describe how can be used to “observe imbalance or limitations”

# Author’s response: Agree. Thanks so much. We have changed it. Yellow in the test.

“To this end, the articular evaluation Range of Motion (ROM) is currently used to ob-serve imbalance or limitations, as possible risk factor or predisposition to suffer future in-jures due to this cause [6,7]. ROM is the maximum angle described between two body segments with a reference plane, which is realised by means of joints. ROM can also be considered as the flexion of the joints or as the degree of muscle contraction. ROM can be used as a reference data to detect muscle asymmetries. ROM can be affected by lack of flexibility, because the muscle chains work less efficiently and the load changes and can lead to injury. In addition, ROM is affected by anatomical, biomechanical, biochemical and neurophysiological factors. In this sense, functional movement tests are currently used by sports scientists for the evaluation of injuries, as they evaluate the mobility and balance and allow us to observe asymmetries or functional imbalance”.

# Editor: Line 69-70. Why SEBT? Is it the gold standard for that? Maybe not. A stronger rationale should be written. Describe the options, and then, move to the SEBT.

# Author’s response:  Agree, good detail. Thanks so much. We have changed it. Yellow in the test.

“This test is considered the gold standard”.

# Editor: Lines 71-82: merely descriptive. Add information about the accuracy, precision, and repeatability of these tests (with values). Additionally, describe the outcomes and how the outcomes have been analyzed in the literature. Add information about the usability of these tests in scientific research.

# Author’s response: Agree, good detail. Thanks so much. We have changed it. Yellow in the test.

“Recently, Cramer et al. [12], based on UQYBT, and with the aim of reducing the ex-penses of adquisition of these tests, established a measurement protocol, Y shaped, 3 laces on the ground and 3 wooden blocks, as the reference of the reached distance, called modi-fied-Upper Quarter Y-Balance Test (mUQYBT). The reliability and concurrent validity of this test was shown with these trials. The results of the study showed a correlation R2=0,96, between the mUQYBT and UQYBT [12]”.

# Editor: Line 83. The introduction to the OB test is not clear. Why this test? What difference in comparison with the others? How he has been used?

# Author’s response: Agree. Thanks so much. We have changed it. Yellow in the test.

“Finally, there is a measurement system, OctoBalance Test (OB), system Check your MotionR, which involves an octagonal platform and 2 measurement systems (retractable steel ruler tapes), being a valid and reliable tool [3] to assess the functional capacities of athletes, to identify weaknesses and asymmetries. In addition, it has been shown as a valid and reliable tool [3] for the measurement of ROM in the lower limb, for which, and up to date as far as we know, there are no previous level studies or evidence which show the use of OB test for the upper limb. Thus, it is necessary its verification and validation for these types of measures in the aforementioned body area”.

# Editor: Line 88: validity? Or concurrent validity?

# Author’s response: Agree. Thanks so much. We have changed it. Yellow in the test.   

“The reliability and concurrent validity of this test was shown with these trials”.

# Editor: Line 91: start the section with an “experimental approach”. Present key elements of study design early in the paper. Describe the setting, locations, and relevant dates, including periods of recruitment, exposure, follow-up, and data collection. Add the timeline of the study.

# Author’s response: Agree. Thanks so much. We have changed it. Yellow in the test.  

The research had an experimental approach. A total of 25 male subjects participated in the study. The OB and the mUQYBT were used as assessment instruments for the study. The assessment process consisted in the performance of different movements, with the right arm and left arm, on the OB and mUQYBT platforms, measuring afterwards the length reached. All the process was repeated a second time, with a week gap between measurements in May 2018. The evaluation was carried out at the laboratory of the European University of the Atlantic, Santander, Spain.

# Editor: Line 93: add information about the “participants”, namely the population, past in sports. Also, add information if they had experienced in the test since this can be a bias for the learning curve.

# Author’s response: Agree, good detail. Thanks so much. We have changed it. Yellow in the test.

“Twenty-five participants participated in the assessment, male athletes, with mean 21.3 ± 2.4 years old, height 176.5 ± 7 cm and body mass 72.96 ± 7.79 kg, being 23 right handed participants and 2 left handed participants, who stated to have an active life, with 11.6 hours of weekly physical activity (Table 1). Before the test, the athletes were familiarised with OB for 3 months”.

# Editor: Line 147: the observers were tested for intra- and inter-observers reliability?

# Author’s response: Agree. Thanks so much. We have changed it. Yellow in the test.  

# Editor: Line 168: for 3 months?

# Author’s response: Agree. Thanks so much. Yellow in the test.  

# Editor: Line 215: Add the outcome extracted and explain how quantitative variables were handled in the analyses.

# Author’s response: Agree. Thanks so much. We have changed it. Yellow in the test.    

Moreover, in order to establish the correlation, the results obtained in the first week were taken into account. The results identified 25 subjects with 73.53 ± 21.22 cm in mUQYBT and 69.54± 16.33 cm in OB, in the medial, superolateral and inferolateral directions, of the right and left arms, in week 1. The assessment of R2, showed almost correlation and an almost linear logarithmic regression, between the scores of OB and mUQYBT, in week 1, with an adjustment value of R2=0.97, for the logarithmic values of the mean.

# Editor: Line 233: details the options to perform the ICC.

# Author’s response: Agree. Thanks so much. We have changed it. Yellow in the test.   

In order to verify the reliability of OB, the ICC was calculated, and ICC was used (3.1) [15]. The ICC (3.1) was used, where each assessor assesses each item, but the assessors are the only assessors of interest. In addition, reliability was calculated from a single meas-urement.

# Editor: Figure 6: add the units in x and y axes. The quality of the figure must be better (resolution).

# Author’s response: Agree. Thanks so much. We have changed it.

# Editor: Table 7. replace “,” by “.” In the sig

# Author’s response: Agree. Thanks so much. We have changed it.

# Editor: Figures 7-12: resolution must be better, the numbers must be bigger and the y and x-axes must have the caption.

# Author’s response: Agree. Thanks so much. We have changed it. Yellow in the test.   

# Editor: Line 306: add the main findings and answers to the research question.

# Author’s response: Agree. Thanks so much. We have changed it. Yellow in the test.   

The aim of this study has been to verify the concurrent validity, reliability and repro-ducibility of OctoBalance Test as a valid and reliable tool to measure articular ROM of the upper limb compared to modified-Upper Quarter Y-Balance Test. We can indicate that the OB is shown as a valid, reliable and reproducible tool for the assessment of the articular ROM in the upper limb, and it could be used for sports scientist for the evaluation of injuries.

# Editor: Line 309: “for prevention of injuries”? “allow observing imbalance or limitations”? This was a study?

# Author’s response: Agree. Thanks so much. We have changed it. Yellow in the test.   

# Editor: Lines 314: It was the aim of the study? Was tested?

# Author’s response: Agree. Thanks so much.

# Editor: Lines 320-322: it was studied? The results were found in this experimental approach? Predicting injury?

# Author’s response: Thanks so much. We have changed it. Yellow in the test.

Reviewer 2 Report

The aim of this study has been to verify the validity, reliability and reproducibility of OctoBalance Test (OB) as a valid and reliable tool to measure articular ROM of the upper limb compared to modified-Upper Quarter Y-Balance Test (mUQYBT).

The authors concluded that OB is shown as a valid, reliable and reproducible tool for the assessment of the articular ROM in the upper limb, and it could be used for the evaluation and prevention of injuries.

The manuscript is interesting and well written. However, some modifications are needed:

Major changes:

Based on the described studies and literature in the Introduction, after the aim of the study, I suggest adding one or two hypothesis that will be tested.

I suggest adding a flow chart of the participants inclusion to the present study.

At the end of the discussion. I suggest adding and discussing some limitations inherent to the present study.

Minor changes:

I suggest that the following part is not needed in the abstract:

“Data are shown as a mean (±) standard deviation (SD). Shapiro-Wilk tests (<50) were applied to verify the normality of data. A significance level p ≤ 0.05 was accepted, with confidence intervals (CI) of 95% in all the measures.”

Please remove.

Introduction. Utilise longer paragraphs; not only one or two sentences.

Table 2 and 3: I suggest changing “ and ‘ to (s) and (min).

Figures 6 to 12. I suggest changing theses figures by clearer ones.

Author Response

# Editor: Based on the described studies and literature in the Introduction, after the aim of the study, I suggest adding one or two hypothesis that will be tested.

# Author’s response: Agree, good detail. Thanks so much. We have changed it. Yellow in the test.

Thus, as a hypothesis of the study, it is proposed that OB is shown as a valid in comparison to the other test, reliable and reproducible tool for the as-sessment of the articular ROM in the upper limb, and it could be used for the evaluation of injuries.

# Editor: I suggest adding a flow chart of the participants inclusion to the present study.

# Author’s response: Thanks so much. Yellow in the test. The description of the participants has been completed.

# Editor: At the end of the discussion. I suggest adding and discussing some limitations inherent to the present study.

# Author’s response:  Thanks so much. Yellow in the test.

The type of sport, the sample and the experience, as it usually happens in these kind of assessments, can be considered as limitations as the election of some implies the rejec-tion of others which could also provide other type of data of wide interest. On the other hand, it had provided a very homogeneous sample, including only men, with a longitu-dinal monitoring for the study and the collection of results.

It should be pointed out that the conclusions provided have been carried out accord-ing to the results obtained in our research, selected under our eligibility criteria, using valid and reliable tools to identify the validity, reliability and reproducibility of the OB as the assessment tool of the articular ROM in the upper limb.

# Editor: I suggest that the following part is not needed in the abstract:

“Data are shown as a mean (±) standard deviation (SD). Shapiro-Wilk tests (<50) were applied to verify the normality of data. A significance level p ≤ 0.05 was accepted, with confidence intervals (CI) of 95% in all the measures.”

# Author’s response: Agree. Thanks so much. We have changed it.

# Editor: Introduction. Utilise longer paragraphs; not only one or two sentences.

# Author’s response: Agree. Thanks so much. We have changed it.

# Editor: Figures 6 to 12. I suggest changing theses figures by clearer ones.

# Author’s response: Agree. Thanks so much. We have changed it.

Reviewer 3 Report

General Comments:

The study Validity, reliability and reproducibility of OctoBalance Test as tool to measure the upper limb compared to modified-Upper Quarter Y-Balance Test is in general an interesting paper. Unfortunately, in the view of the reviewer, the study does not fit the aims and scopes of IJERPH. That been said, this is my opinion. If you strongly disagree, I encourage you to write to the editor and argue for you fit. I know this is disappointment for you, but remember, you can publish the study maybe in another MDPI journal, such as Sports, where a higher fit to the aims and scopes is expected. However, the study should be significantly revised before resubmission. You will find advice on this below and under the specific comments:

Unfortunately, the introduction does not lead stringently to the research question and does not set out its relevance. Similarly, the method should be more detailed in parts, please see the specific comments. The quality of the presentation of results should be fundamentally revised. Additionally, the discussion does not read like a real discussion, it is rather a repetition of results for the most part.

Specific comments:

Line 95: How was the active lifestyle and the amount of exercise per week determined? Please name the standardised survey instrument. Was there a breakdown of the types of sports the respondents participated in?

Line 97: The table appears redundant

Line 102: How were the inclusion criteria of the subjects checked? Questionnaire/examination doctor?

Line 129: Please explain why the OctoBalance test should be validated at the UQYBT. Is this the gold standard? If yes, please also state the quality criteria.

Line 147: Was the rater agreement evaluated?

Line 232 : Please calculate absolute reliability (typicalerror of measurement expressed as coefficient of variation)

Line 254: The table appears redundant

Line 260: The table appears redundant

Line 266: The table appears redundant

Line 273: The image seems to be of poor quality

Line 281: please add calculation of  coefficient of variation

Line 281: Please add Information to single measure and mean measures

Line 295: The image 7 -12 seems to be of poor quality

Author Response

# Editor: Line 95: How was the active lifestyle and the amount of exercise per week determined? Please name the standardised survey instrument. Was there a breakdown of the types of sports the respondents participated in?

# Author’s response: Agree. Thanks so much. We have changed it.

Twenty-five participants participated in the assessment, male athletes, with mean 21.3 ± 2.4 years old, height 176.5 ± 7 cm and body mass 72.96 ± 7.79 kg, being 23 right handed participants and 2 left handed participants, who stated to have an active life, with 11.6 hours of weekly physical activity (Table 1) For the evaluation of physical activity the In-ternational Physical Activity Questionnaires (IPAQ) was used. Before the test, the athletes were familiarised with OB for 3 months.

# Editor: Line 129: Please explain why the OctoBalance test should be validated at the UQYBT. Is this the gold standard? If yes, please also state the quality criteria.

# Author’s response: Agree, good detail. Thanks so much. We have changed it. Yellow in the test.

The OB was used as assessment tool, system Check your MotionR (Figure 1), which involves an octagonal platform and 2 measurement systems (retractable steel ruler tapes), being a valid and reliable tool [3] to assess the functional capacities of athletes, to identify weaknesses and asymmetries and to provide a continuous feedback during the practice of the corrective exercises.

OB has been shown as a valid and reliable tool [3] for the measurement of ROM in the lower limb, for which, and up to date as far as we know, there are no previous level stud-ies or evidence which show the use of OB test for the upper limb. Thus, it is necessary its verification and validation for these types of measures in the aforementioned body area compared to the mUQYBT. This test is considered the gold standard.

# Editor: Line 147: Was the rater agreement evaluated?

# Author’s response: Agree. Thanks so much. We have changed it.

# Editor: Line 232: Please calculate absolute reliability (typicalerror of measurement expressed as coefficient of variation)

# Author’s response: Agree. Thanks so much. We have changed it. Yellow in the test.

# Editor: Line 273: The image seems to be of poor quality

# Author’s response: Thanks so much. We have changed it.

# Editor: Line 281: please add calculation of coefficient of variation

# Author’s response: Agree. Thanks so much. We have changed it. Yellow in the test.

# Editor: Line 281: Please add Information to single measure and mean measures

# Author’s response: Thanks so much. We have changed it.

# Editor: Line 295: The image 7 -12 seems to be of poor quality

# Author’s response: Thanks so much. We have changed it.

Round 2

Reviewer 1 Report

The authors made a meaningful effort to change and improve the article. I think that can be accepted in the present form.

Author Response

Dear Reviewer:

Thanks so much

Reviewer 2 Report

I suggest that this version is suitable for publication.

Author Response

Dear Reviewer

Thanks so much

Reviewer 3 Report

Many thanks to the authors for the revision of the manuscript. From my point of view, the revision was beneficial for the quality of the manuscript.  

I have only one further comment of a more editorial nature: the quality of the illustrations could be in parts further improved. 

Author Response

Dear Reviewer:

Thanks so much. We have considerd your suggestions. Attached the final version of the text

King Regards
